# The Diversified Ensemble Neural Network

**Shaofeng Zhang**[1] **, Meng Liu**[1] **, Junchi Yan**[2]*
[1] University of Electronic Science and Technology of China
[2] Department of CSE, and MoE Key Lab of Artificial Intelligence, AI Institute
Shanghai Jiao Tong University
{sfzhang,2017221004027}@std.uestc.edu.cn, yanjunchi@sjtu.edu.cn

## Abstract

Ensemble is a general way of improving the accuracy and stability of learning models, especially for the generalization ability on small datasets. Compared with tree-based methods, relatively less works have been devoted to an in-depth study on effective ensemble design for neural networks. In this paper, we propose a principled ensemble technique by constructing the so-called diversified ensemble layer to combine multiple networks as individual modules. Through comprehensive theoretical analysis, we show that each individual model in our ensemble layer corresponds to weights in the ensemble layer optimized in different directions. Meanwhile, the devised ensemble layer can be readily integrated into popular neural architectures, including CNNs, RNNs, and GCNs. Extensive experiments are conducted on public tabular datasets, images, and texts. By adopting weight sharing approach, the results show our method can notably improve the accuracy and stability of the original neural networks with ignorable extra time and space overhead.

## 1 Introduction

Deep neural networks (DNNs) have shown expressive representation power based on the cascading structure. However, their high model capacity also leads to the overfitting issue and making DNNs a less popular choice on small datasets, especially compared with decision tree-based methods.

In particular, ensemble has been a de facto engineering protocol for more stable prediction, by combining the outputs of multiple modules. In ensemble learning, it is desirable that the modules can be complementary to each other, and module diversity has been a direct pursuit for this purpose. In tree-based methods such as LightGBM [1] and XGBoost [2], diversity can be effectively achieved by different sampling and boosting techniques. However, such strategies are not so popular for neural networks, and the reasons may include: i) neural networks (and their ensemble) are less efficient; ii) the down-sampling strategy may not work well on neural networks as each of them can be more prone to overfitting (e.g., by using only part of the training dataset), which affects the overall performance. In contrast, decision tree models are known more robust to overfitting, and also more efficient.

We are aimed to devise a neural network based ensemble model to be computationally efficient and stable. In particular, the individual models is trained for maximizing their diversity such that the ensemble can be less prone to overfitting. To this end, we propose the so-called diversified ensemble layer, which can be used as a plug in with different popular network architectures, including CNNs [3], RNNs [4], and GCNs [5]. Meanwhile, due to its partial weight sharing strategy, it incurs relatively small extra time overhead in both training and inference. The main contributions are as follows:

**1)** Instead of adopting existing popular down-sampling and feature selection strategies, we propose another principled technique, whereby each individual model can use full features and samples for

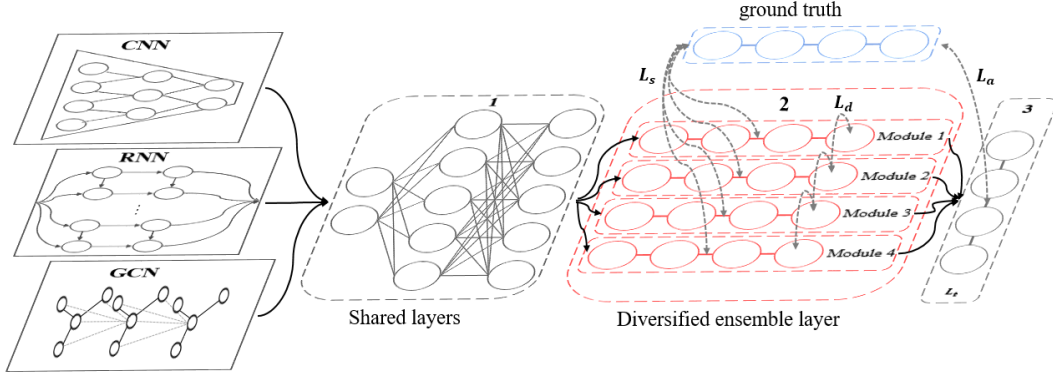

Figure 1: The ensemble network with the proposed diversified ensemble layer (in red). The outputs of the front-end network, as can be embodied by architectures like CNN, RNN, GCN is fed into the FC layer to extract features. Until this step, all the weights are shared across different modules in the ensemble layer. The modules are trained together with the other parts of the whole network.

end-to-end learning. Thus, the individual models can be optimized in different directions for diversity, to enhance the generalization ability. We further provide theoretical analysis to show its effectiveness.

**2)** We propose a novel and adaptive learning procedure, which balances model diversity and training accuracy, so as to improve its generalization ability on testing data. Its efficiency is fulfilled by partial weight sharing across individual modules, which also plays a role in extracting common features for further extraction by the individual modules.

**3)** Extensive experimental results show that our ensemble layer can significantly improve the accuracy by taking relatively low extra time and space, as shown in our extensive experimental results. Our ensemble layer can also be easily applied to CNNs, RNNs, GCNs, etc.

## 2   Related Works

We discuss two areas closely related to our works: weight sharing in neural networks and diversity learning. Readers are referred to [6, 7] for a comprehensive review on ensemble learning.

**Weight sharing.** The work ENAS [8] presents a NAS training scheme with weight sharing (WS), which measures the performance of architecture with the weights inherited from the trained supernet. Since then, weight sharing has been widely adopted to exploit NAS in various applications, such as network compression [9], objection detection [10, 11]. Besides, the work [12] adopts weight sharing strategy for unsupervised neural machine translation. It shares the weights of the last few layers of two encoders and the first few layers of two decoders, and adversarial technique is employed to strengthen the shared potential space. WSMS-Net [13] proposes a new WS strategy, which shares parameters in the front-back direction of images in addition to the ordinary CNNs. Unlike existing WS tactics, in this paper, we employ a full-weight sharing approach. As shown in Fig. 1, for each individual model, all the weights are the same except for the last few layers.

**Diversity learning.** In general, the diversity of individual modules can improve ensemble's generalization ability and stability. Random Forest [14] adopts the down-sampling strategy, which utilizes bootstrapping to select samples and features for training to increase the diversity of different decision trees. The work [15] proves encouraging high diversity among individual classifiers will reduce the hypothesis space complexity of voting, and thus better generalization performance can be expected. A number of diversity metrics are devised in [16], providing a framework to select individual models. More recently, it is proposed to increase structural diversity of decision tree [17], to enhance the performance on small tabular datasets. GASEN is devised in [18], which proposes a new model selective strategy with far smaller size but stronger generalization ability than bagging and blending all individual models. Compared with GASEN, the proposed ensemble layer in this paper focuses on how to construct high diversity individual models, while GASEN focuses on how to select models to aggregate. XGBoost [2] and LightGBM [1] are developed based on gradient boosting decision tree

(GBDT) [7], whereby XGBoost utilizes the the second-order Taylor series and regularization term for a better generalization. While LightGBM adopts exclusive feature bundling and gradient-based one-side sampling to speed up. GrowNet [19] is proposed, which adopts a gradient boosting method to neural networks. DBD-CENet [20] also adopts boosting strategy to neural networks, besides, they combines the idea of knowledge distillation (teacher-student network) and co-training. In detail, DBD-CENet uses one network to estimate the residual of the other network iteratively. Finally, they fine-tune the two branches of DBD-CENet in an iterative way with every epoch. ADPNet [21] brings diversity to adversarial defense. Ensemble enhanced by diversity can greatly improve the robustness of models. ADPNet's diversity is fulfilled by dot product in embedding layer, which limits its applicability to tabular data and regression task. In these ensemble methods, individual models are trained independently, and the diversity is often fulfilled by feature and sample down-sampling, rather than in a joint learning fashion. There are very few methods for end-to-end ensemble learning whereby the diversity of each module is jointly modeled. This paper aims to fill this gap.

## 3 The Proposed Diversified Ensemble Neural Network

### 3.1 Architecture and Objective Design

Without loss of generality, consider the network for binary classification. Given a set of data $\{\mathbf{X}, \mathbf{Y}\}$, where $\mathbf{Y}$ is the label in $\{-1, +1\}$. The objective $\mathcal{L}_t$ can be generalized as a composite function:

$$\mathcal{L}_t = \mathcal{H}(q(\mathbf{W}_H \cdot a(\mathbf{W}_{H-1} \cdots a(\mathbf{W}_1 \cdot \mathcal{T}(\mathbf{X})))), \mathbf{Y}) \quad (1)$$

where $q(\cdot)$ is a normalized function, $\mathcal{H}$ is the cross-entropy loss, $\mathbf{W}_H$ denotes the parameter matrix connecting layer $H-1$ and layer $H$, $a$ is activation and $\mathcal{T}(\cdot)$ is the feature extractor e.g. CNN, GCN.

Specifically, our proposed ensemble network is designated as shown in Fig. 1, which consists of (from left to right): i) the application dependent part in the form of either CNN, GCN or RNN; ii) the shared fully connected layers; iii) the proposed diversified ensemble layer that is comprised of multiple modules; iv) the final scoring layer (e.g. classification, regression etc). Overall, the devised loss contains three parts (see Fig. 1):

1) cross-entropy between $\mathbf{Y}$ and individual modules, as written by:

$$\mathcal{L}_s = \mathcal{H}(q(\mathbf{W}_{H-1}^{(i)} \cdots \sigma(\mathbf{W}_1 \cdot \mathcal{T}(\mathbf{X}))), \mathbf{Y}) \quad (2)$$

where $\mathbf{W}_{H-1}^{(i)}$ denotes the parameter connecting layer $H-2$ and the $i$-th neuron in layer $H-1$.

2) diversity of individual modules, which is a common metric in the work [15], can be expressed as:

$$\mathcal{L}_d = 1 - \frac{1}{N} \sum_{1 \leq i \neq j \leq N} q(\mathbf{W}_{H-1}^{(i)} \mathbf{X}_{H-1}) q(\mathbf{W}_{H-1}^{(j)} \mathbf{X}_{H-1}) \quad (3)$$

where $\mathbf{X}_{H-1}$ is the input of layer $H-1$. For regression, the loss can be quantified as:

$$\mathcal{L}_d = \frac{1}{N} \sum_{1 \leq i \neq j \leq N} \left( \mathbf{W}_{H-1}^{(i)} \mathbf{X}_{H-1} - \mathbf{W}_{H-1}^{(j)} \mathbf{X}_{H-1} \right)^2 . \quad (4)$$

3) aggregated loss as written as follows, given $\mathbf{Y}^{(i)} = q(\mathbf{W}_{H-1}^{(i)} \mathbf{X}_{H-1})$ as the individual output:

$$\mathcal{L}_a = \mathcal{H}\left( \sum_i^N \gamma_i \cdot \mathbf{Y}^{(i)}, \mathbf{Y} \right) \quad (5)$$

where $\gamma_i$ represents the aggregate weight of the last layer, which is bounded by $\sum_i^N \gamma_i = 1$. Note that while updating $\gamma_i$, other layers' parameters will be frozen. Thus, the total loss is given by:

$$\mathcal{L}^{(i)} = \mathcal{L}_s + \mathcal{L}_a - \alpha^{(i)} \cdot \mathcal{L}_d \quad (6)$$

where $\mathcal{L}^{(i)}$ is the total loss in the $i$-th iteration, and $\alpha^{(i)}$ is a shrink parameter which can be adaptively updated by sigmoid function $\alpha^{(i)} = \sigma(\mathbb{E}_{\mathbf{W}}(|\nabla_{\mathbf{W}} \mathcal{L}_s^{(i-1)}|))$. The reason for this design is that in training, the gradient of $\mathcal{L}_s$ will continue to decrease, while $\mathcal{L}_d$ will grow with the increase of the variance of neuron's output in the ensemble layer. So the mean of the gradient of $\mathcal{L}_s$ can be used to balance $\mathcal{L}_s$ and $\mathcal{L}_d$. The algorithm for regression is given in Alg. 1 (similar for classification).

---
**Algorithm 1** Diversified Ensemble Layer for Regression Network Training and Prediction
---
1: **Input**: number individual model $N$, data set $\{\mathbf{X}, \mathbf{Y}\}$, $max\_iter$
2: **Initialization**: shared fully-connected weights $\mathbf{W}$, feature extractor $\mathcal{T}$, individual weights $\gamma$, $epoch = 0$, $i = 0$
3: **Output**: parameters and prediction of ensemble model
4: **while** $epoch < maxiter$ **do**
5:   **while** $i < N$ **do**
6:     Forward propagation of part I in Eq. 1: $\mathbf{Y}^{(i)} \leftarrow \mathbf{W}_{H-1}^{(i)} \cdots \sigma(\mathbf{W}_1 \cdot \mathcal{T}(\mathbf{X}))$;
7:     Update $\mathcal{L}_s \leftarrow (\mathbf{Y}^{(i)} - \mathbf{Y})^2$ by Eq. 2;
8:     Update $\mathcal{L}_d \leftarrow \sum_i \sum_j (\mathbf{Y}^{(i)} - \mathbf{Y}^{(j)})^2$ by Eq. 3;
9:     Update $\mathcal{T} \leftarrow \nabla_\mathcal{T} \mathcal{L}_s(\mathbf{Y}^{(i)}, \mathbf{Y}) - \alpha \cdot \nabla_\mathcal{T} \mathcal{L}_d(\mathbf{Y}^{(i)}, \mathbf{Y}^{((j)})$ by Eq. 2 and Eq. 3;
10:    Update $\mathbf{W} \leftarrow \nabla_\mathbf{W} \mathcal{L}_s(\mathbf{Y}^{(i)}, \mathbf{Y}) - \alpha \cdot \nabla_\mathbf{W} \mathcal{L}_d(\mathbf{Y}^{(i)}, \mathbf{Y}^{((j)})$ by Eq. 2 and Eq. 3;
11:    Update $\alpha \leftarrow \sigma(\mathbb{E}(|\nabla_\mathbf{W} \mathcal{L}_s(\mathbf{Y}^{(i)}, \mathbf{Y})|))$;
12:  **end while**
13:  Update $\mathbf{Y}_p \leftarrow \sum_i \gamma_i \cdot \mathbf{Y}^{(i)}$, $\mathcal{L}_t \leftarrow (\mathbf{Y} - \mathbf{Y}_p)^2$ in Eq. 5;
14:  Update $\gamma_i \leftarrow \gamma_i - \nabla_{\gamma_i} \mathcal{L}_t(\mathbf{Y}_p, \mathbf{Y})$ by Eq. 5 followed by normalization $\gamma_i \leftarrow \frac{e^{\gamma_i}}{\sum e^{\gamma_j}}$;
15: **end while**
16: **return** parameters $(\mathcal{T}, \mathbf{W}, \gamma)$, prediction $\sum_i \gamma_i \cdot \mathbf{Y}^{(i)}$;
---

## 3.2 Theoretical Analysis

**Theorem 1** *The loss $\mathcal{L}_s$ and $\mathcal{L}_d$ **can be simultaneously optimized.** Given a set of linear mapping functions $\mathcal{H} = \{f_i\}_{i=1}^N$ and a set of linearly distributed samples $\mathbf{X}, \mathbf{Y}$ in regression setting. Let $\mathcal{L}_i = \mathcal{H}(f_i(\mathbf{X}, \theta_i), \mathbf{Y})$ and $div(i,j) = \mathcal{H}(f_i(\mathbf{X}, \theta_i), f_j(\mathbf{X}, \theta_j))$, where $i \neq j$ and $\mathcal{H}$ indicates the mean square loss. Then there always exists an optimal direction, which makes $f_i(\mathbf{X}, \theta_i)$ decrease $\mathcal{L}_i$ while $div(i,j)$ increase.*

**Proof** *Suppose there is a perfect mapping function $f_r$, which can make $f_r(\mathbf{X}; \theta_r) = \mathbf{Y}$. Then the gradient of $\mathcal{L}_i$ can be written as:*

$$\frac{\partial \mathcal{L}_i}{\partial \theta_i} = \frac{[f_i(\mathbf{X}) - f_r(\mathbf{X})]^\top [f_i(\mathbf{X}) - f_r(\mathbf{X})]}{\partial f_i(\mathbf{X}; \theta_i)}$$

*where $\theta_i$ is the parameter in $f_i$. The gradient of individual diversity loss $div(i,j)$ can be written as $\partial\{div(i,j)\}/\partial\theta_i = [f_i(\mathbf{X}) - f_r(\mathbf{X})]^\top [f_i(\mathbf{X}) - f_r(\mathbf{X})]/\partial\theta_i$.*

*To prove the direction of gradient, it's a common approach to measure the loss angle between two gradient in [22]. Since we want to maximize the individual diversity, the loss angle can be written as:*

$$-\frac{\partial div(i,j)}{\partial \theta_i} \cdot \frac{\partial \mathcal{L}_s}{\partial \theta_i} = 4 \cdot (\mathbf{X}^\top \mathbf{X}\theta_i - \mathbf{X}^\top \mathbf{X}\theta_r)^\top (\mathbf{X}^\top \mathbf{X}\theta_j - \mathbf{X}^\top \mathbf{X}\theta_i),$$

*which can obtain the maximum when $\theta_i = (\theta_j + \theta_r)/2$. And the maximum value is $(\mathbf{X}^\top \mathbf{X}(\theta_j - \theta_r))^\top (\mathbf{X}^\top \mathbf{X}(\theta_j - \theta_r)) \geq 0$. Since angles between two gradients can be less than 90 degrees, $f_i(\mathbf{X}, \theta_i)$ can search the direction, which minimizes $\mathcal{L}_i$ and meanwhile maximizes $div(i,j)$.* $\square$

The above proof is based on the assumption that samples are in linear distribution, while the real data can be non-linear. Fortunately, compared with linear regression and Logistic regression, the advantage of neural networks is that the feature extractor e.g., CNNs, GCNs and activation function of neural networks can enforce non-linear transformation. Given a non-linear distribution of data $(\mathbf{X}, \mathbf{Y})$, it can be spread forward simply according to Eq. 2. Finally, we can utilize the final representation as a linearly distributed data, which is able to be separated by a linear model.

**Theorem 2** *Generalization improvement enhanced by diversity. Given a set of binary classification data set $\mathcal{D} = \{\mathbf{X}, \mathbf{Y}\}_{i=1}^m$ sampled in the distribution $\mathcal{U}$ and classifier $\mathcal{H} = \{f_i(\mathbf{x})\}_{i=1}^N$ to map feature space $\mathbf{X}$ to label space $\mathbf{Y} = \{-1, +1\}$ and the diversity function is given by:*

$$div(\mathcal{H}) = 1 - \frac{1}{N} \sum_{1 \leq i \neq j \leq N} \frac{1}{m} \sum_{k=1}^m f_i(\mathbf{x}_k) f_j(\mathbf{x}_k).$$

With the probability at least $1 - \delta$, for any $\theta > 0$, the generalization error can be bounded by:

$$err_{\mathcal{U}}(f) \leq err_{\mathcal{D}}(f) + \frac{C}{\sqrt{m}} \sqrt{\frac{\ln N \ln(m\sqrt{1/N + (1 - 1/N)(1 - div(\mathcal{H}))})}{\theta^2} + \ln \frac{1}{\delta}} \quad (7)$$

where $C$ is a constant.

**Proof** *The average of classifiers is given by $f(\mathbf{x}; \mathcal{H}) = \frac{1}{N} \sum_{i=1}^{N} f_i(\mathbf{x})$. Then $\|f(\mathbf{x}; \mathcal{H})\|_2^2$ becomes:*

$$\|f\|_2^2 = \sum_{i=1}^{m} \left( \frac{1}{N} + \frac{1}{N^2} \sum_{1 \leq i \neq j \leq N} f_j(\mathbf{x}_i) f_k(\mathbf{x}_i) \right) = m \left( \frac{1}{N} + (1 - div(\mathcal{H}))(1 - \frac{1}{N}) \right) \geq 0$$

*While $\|f\|_2^2$ is always non-negative. Then $\|f\|_1$ can be obtained by:*

$$\|f\|_1 = \sqrt{m}\|f\|_2 = m\sqrt{1/N + (1 - 1/N)(1 - div(\mathcal{H}))}$$

*Then, we adopt the proof strategy in the work [22]. Firstly, divide the interval $[-1 - \epsilon/2, 1 + \epsilon/2]$ to $[4/\epsilon + 2]$ sub-intervals and each of the sub-interval's size is no larger than $\epsilon/2$. Let $-1 - \epsilon/2 = \theta_0 < \theta_1 < \cdots < \theta_m = 1 + \epsilon/2$ be the boundaries of the intervals.*

*Then, we use $j_l(i)$ to represent the maximum index of $\theta_i$ such that $f_i(\mathbf{x}) - j_l(i) \geq \epsilon/2$ and use $j_r(i)$ represent the minimum index of $\theta_i$ such that $f_i(\mathbf{x}) - j_r(i) \leq -\epsilon/2$. Then, we let $f_i^{(1)} = [f_i - j_l(i)]$ and $f_i^{(2)} = [-f_i + j_r(i)]$. Similar to [22], which constructs the relation between above indexes, here we construct a pair of $(f_i^{(1)}, f_i^{(2)})$. Let $f_p(x) = p \cdot sign(x)|x|^{p-1}$. We can define:*

$$\mathcal{G} = (f_i^{(1)}, f_i^{(2)}) = f_p \left( \sum_{i=1}^{m} \alpha_i f_i^{(1)} + \sum_{i=1}^{m} \beta_i f_i^{(2)} \right) \quad s.t. \quad \sum_{i=1}^{m}(\alpha_i + \beta_i) \leq 36(1 + \ln N)/\epsilon^2.$$

*where $\alpha_i$ and $\beta_i$ are both non-negative. It can be easily seen that the covering number $\mathcal{N}_{\infty}(\mathcal{H}, \epsilon, m)$ is no more than the number of possible $\mathcal{G}$ constructed above. Take $\|f\|_1 = m\sqrt{1/N + (1 - 1/N)(1 - div(\mathcal{H}))}$ to the above equation, we can get that the number of possible valus of $f_i^{(1)}$ is no more than $m\lceil 4\sqrt{1/N + (1 - 1/N)(1 - div(\mathcal{H}))}/\epsilon + 2\rceil$. The possible value of $\mathcal{G}$ is upper-bounded by:*

$$\mathcal{N}_{\infty}(\mathcal{H}, \epsilon, \mathcal{D}) = \left( 2m\lceil 4\sqrt{1/N + (1 - 1/N)(1 - div(\mathcal{H}))}/\epsilon + 2\rceil + 1 \right)^{36(1+\ln N)/\epsilon^2}.$$

*From the Lemma 4 in the work [23], we can get:*

$$err_{\mathcal{U}}(f) \leq err_{\mathcal{D}}(f) + \sqrt{\frac{2}{m} \ln(\mathcal{N}_{\infty}(\mathcal{H}, \epsilon/2, 2m) + 2)/\delta}.$$

*Finally, take the $\mathcal{N}_{\infty}(\mathcal{H}, \epsilon, \mathcal{D})$ to the above equation, we can complete the proof.* □

**Theorem 3** *Error reduction by aggregated based ensemble. Given a set of data samples $\mathcal{D} = \{\mathbf{X}, \mathbf{Y}\}_{i=1}^{m}$ and a set of predictor $\mathcal{H} = \{f_i\}_{i=1}^{N}$, ensemble can reduce the $err_{\mathcal{D}}$ of the predictor.*

**Proof** *We discuss the cases for regression and classification, respectively.*

*For regression task, take $(x, y)$ as the input of a regressor, then the expectation of overall regression squared error can be written by:*

$$err_{\mathcal{D}} = \mathbb{E}_{\mathcal{D}} \left( \mathbf{Y} - \frac{1}{N} \sum_{i=1}^{N} f_i(\mathbf{X}) \right)^2 = \frac{1}{N} \sum_{i=1}^{N} (\mathbf{Y} - f_i(\mathbf{X}))^2 - \sum_{i=1}^{N} \left( \frac{1}{N^2} \sum_{i=1}^{N} f_i(\mathbf{X}) - \frac{1}{N} f_i(\mathbf{X}) \right)^2$$
$$(8)$$

*where the second term R.H.S. implies the diversity of individual predictor, which is always non-negative. Thus, the bagging predictor can improve the accuracy on $\mathcal{D}$ in regression task.*

*For classification task, according to Chapter 4.2 in the work [6], we can derive:*

$$f_i(\mathbf{X}) = \sum_j Q(j|\mathbf{x})P(j|\mathbf{x})$$
$$(9)$$
$$\mathcal{F}(\mathbf{X}) = \int_{\mathbf{x} \in \mathcal{D}} \max_j P(j|\mathbf{x})P_{\mathbf{X}}dx + \int_{\mathbf{x} \in \mathcal{D}} \left[ \sum_j I\left(\mathcal{H}(\mathbf{x}) = j\right) P(j|\mathbf{x}) \right] P_{\mathbf{X}}(dx).$$

*where $P_{\mathbf{X}}(\mathbf{x})$ denotes the distribution of $\mathbf{X}$, $Q(j|\mathbf{x})$ represents the relative frequency of class label $j$ predicted by $f(\mathbf{x})$ with input $\mathbf{x}$, and $P(j|\mathbf{x})$ denotes the conditional probability of sample $\mathbf{x}$. The highest accuracy of $\mathcal{F}(\mathbf{X})$ is $s = \int \max_j P(j|\mathbf{x})P_{\mathbf{X}}(dx)$. The sum $\sum_j Q(j|\mathbf{x})P(j|\mathbf{x})$ is far less than $P(j|\mathbf{x})$. Thus, the individual classifier can be far from optimal, while the aggregated predictor $\mathcal{F}$ is nearly optimal.*

*Based on the above derivation and analysis, we have proven that on both regression (in Eq. 8) and classification (in Eq. 9) tasks, ensemble can reduce $err_{\mathcal{D}}$.* $\square$

From the above proof, we can conclude that since our ensemble layer can reduce $err_{\mathcal{D}}$ while increases the diversity of individual models, then the ensemble model has a smaller generalization error $err_{\mathcal{U}}$.

**Efficiency analysis.** Assume a neural network (NN) in study has $\#hidden \times \#node$ parameters in total, *Ens-NN* (random NN aggregate) will optimize $\#hidden \times \#node \times \#model + \#model$ parameters, where $\#hidden$ is the number of hidden layers, $\#node$ is the number of neurons in a hidden layer, and $\#model$ is the amount of individual model. While in the ensemble layer, only about $\#node \times \#model + \#model$ more parameters needs to be optimized, where $1 << \#hidden$. Then we significantly reduce the training space consumption with greatly improving the accuracy.

Back-propagation is time-consuming. Suppose $\#model$ represents the total number of individual models, while the time required for forward computing loss $\mathcal{L}_a$ is denoted by $\#single$, and the time required for a back-propagation is $\#bp$. In an iteration, the time required for a individual network is $\#bp + \#single$. It takes about $\#bp + \#model \times \#single$ after adding the ensemble layer. While traditional ensemble method needs to train $\#model$ models, whose time consumption is $\#model(\#bp + \#single)$. The time consumption of $\#single$ is far less than that of $\#bp$, especially in very deep neural networks. Thus, our proposed ensemble layer can greatly improve the accuracy with ignorable extra time and space consumption.

## 3.3 Remarks

Note that though the neural network involves a non-convex function, the last layer is often set a linear regression or linear classification, which is convex. Unfortunately, we cannot only consider the parameters in the last linear layer. Because the output of the previous layers changes with the gradient. Even if the input is fixed, and the input of the last layer is a matrix of shape $nP$. If the matrix $(\mathbf{X}^{\top}\mathbf{X})$ is invertible, then the optimal parameter $\mathbf{W} = (\mathbf{X}^{\top}\mathbf{X})^{-1}\mathbf{X}^{\top}\mathbf{Y}$ can be obtained directly by the least square method. If $(\mathbf{X}^{\top}\mathbf{X})$ is not reversible, the optimal value of $\mathbf{W} = (\mathbf{X}^{\top}\mathbf{X} + \lambda I)^{-1}\mathbf{X}^{\top}\mathbf{Y}$ can be obtained by ridge regression. However, the time complexity of solving the inverse matrix is $O(nP^2) \approx O(n^3)$. When the amount of data is huge, it will be difficult to obtain. Therefore, a more extensive approach is often to use the gradient descent method to optimize the solution gradually.

Most existing ensemble methods [24, 25, 26] follow a two-stage process: i) building individual modules; ii) aggregation of outputs of modules. The individual modules are mostly trained independently without interaction. In contrast, we aim to jointly train the predictors by maximizing their diversity $\mathcal{L}_d$ in Eq. 3 and minimizing their own prediction loss $\mathcal{L}_s$ in Eq. 2. Besides, weight sharing is adopted in fully-connected layers, which reduces the time and space complexity.

## 4 Experiments

Experiments are performed on tabular datasets with NN, images with CNN, and texts with RNN and GCN. For tabular datasets, we mainly use Higgs-Boson, KDD10, and Credit and compare it with some boosting models. For image datasets, we run experiments on CIFAR-10 and CIFAR-100 and compare it with vanilla CNN. For texts, we mostly test text classification, and the ensemble layer is added after GRU and LSTM. For the hyper-parameter initialization, $\gamma_i$ are initialized to $1/N$ and the hyper-parameter $\alpha$ are initialized to 1.And the training procedure of the Ensemble layer consists of two training stages. Firstly, we optimize W by each single loss $\mathcal{L}_s$ in Eq. 2 and $\mathcal{L}_d$ in Eq. 3. Then, we fix W and optimize by Lt by Eq. 5. The number of individual models is set $N = 4$ universally. All the experiments are implemented using PyTorch on a single NVIDIA 1080Ti GPU.

Table 1: Description of the datasets used in the experiments.

| dataset | #data | #feature | Description | Task | Metric |
|---|---|---|---|---|---|
| Higgs Boson | 10M | 28 | Dense | Classification | AUC |
| KDD10 | 19M | 29M | Sparse | Binary classification | AUC |
| Credit | 7M | 107 | Dense & Sparse | Regression | MAE |
| CIFAR-10/100 | 60k | 32*32 | Image | Classification | Accuracy |

| dataset | #Train | #Test | #Words | #Classes | Ave. Length |
|---|---|---|---|---|---|
| 20NG | 11,314 | 7,532 | 42K | 20 | 221 |
| R8 | 5,485 | 2,189 | 7K | 8 | 65 |
| R52 | 6,532 | 2,568 | 9K | 52 | 70 |
| Ohsumed | 3,357 | 4,043 | 14K | 23 | 135 |
| MR | 7,108 | 3,554 | 18K | 2 | 20 |
| SST | 8,544 | 3,311 | 18K | 5 | 18 |

## 4.1 Dataset and Implementation Details

**Datasets.** The first dataset is the Higgs boson [27] dataset from high energy physics. To compare XGBoost and LightGBM, we randomly select 10M instances as training set and use the rest as testing set. The second dataset is KDD10 [28], which contains many sparse features. As the neural networks usually perform well on the sparse feature, we can directly use these features. The third dataset Credit [29]. It contains both Dense and Sparse features. Due to the Dense feature has discrete (jobs et al.) and continuous features (age etc.), we transform discrete features to one-hot encoding.

The image datasets are CIFAR-10 and CIFAR-100, whose images are both of $32\times32$. Before training, we resize them into $256\times256$. Then we use the pre-trained VGGs or ResNets model as CNN part. CNNs can be used as a feature extractor to do non-linear transformation. Notably, the feature extracting and ensemble layer training is in one step.

For text datasets, The 20NG [30] contains 18,846 documents totally categorized into 20 newsgroup documents. Ohsumed corpus [31] contains 7,400 documents, which is divided into 23 categories. R52 and R8 [32] are two subsets of the Reuters 21578 dataset. R8 has eight categories and is split to 5,485 training and 2,189 test documents. R52 has 52 categories and is split to 6,532 training and 2,568 test documents. MR [33] corpus has 5,331 positive and 5,331 negative reviews. SST [34] contains five categories, which mainly classify the sentiment of movie reviews. Table 1 in shows the profile of used datasets.

**Baselines.** To our knowledge, there are only a few ensemble works about neural networks or neural layers. To prove our ensemble layer has high accuracy on regression and classification tasks, we compared our model with the state-of-the-art in these domains. For the tabular datasets, it is well known that boosting tree models such as LightGBM and XGBoost has higher accuracy. We mainly compared accuracy with them. For the image datasets, most works are related to CNNs. We mostly compare the ensemble layer with vanilla CNNs on accuracy, time spending, and space spending. For the text datasets, we mainly do ablation experiments and adopt TF-IDF + LR as baselines. For RNN models, we adopt RNNs' output as embedded features. For instance, $N = 4$ means taking the last four hidden outputs as the input of the ensemble layer. For GCN and Text-GCN, we use two graph convolutional layers as feature extractors. The ensemble layer is added after the graph layers.

## 4.2 Experimental Results

**Results on tabular datasets.** Part I of Table 2 shows AUC and MAE on tabular and image datasets. Our ensemble model outperforms on the KDD10 and Credit dataset. However, it performs slightly worse than LightGBM on the Higgs Boson dataset. We conjecture this is because the Higgs Boson dataset is dense, which is more friendly to the branching operation used in boosting tree models. While on sparse datasets, neural network can perform slightly better.

**Results on image datasets.** Part II of Table 2 shows top-1 and top-5 accuracy score on CIFAR datasets. Both VGG-16 and VGG-19 models performed much better after adding our ensemble layer (Note VGG-19 has three more convolutional layers than VGG-16). Although random ensemble can improve accuracy slightly, it will spend much more time on training and ensemble, especially in such deep networks.

Table 2: Evaluation on tabular and images datasets. NN: individual network, R-Forest: random forest, Ens-NN: random neural network ensemble, DEns-NN: NN+Ensemble layer. Best result in bold.

| Part **I** (tabular data) | R-Forest | LGBM | XGBoost | NN | Ens-NN | DEns-NN |
|---|---|---|---|---|---|---|
| Higgs Boson (accuracy) | 82.26% | **83.10%** | 83.04% | 82.83% | 82.89% | 83.08% |
| KDD10 (accuracy) | 76.88% | 78.74% | 77.96% | 77.14% | 77.23% | **79.19%** |
| Credit (MAE) | 16.6413 | 14.5520 | 14.6702 | 14.9901 | 14.9864 | **14.1028** |
| Part **II** (image data) | VGG16 | Ens-VGG16 | DEns-VGG16 | VGG19 | Ens-VGG19 | DEns-VGG19 |
| CIFAR-10 (Top-1 acc.) | 93.56% | 93.98% | 94.89% | 93.71% | 93.80% | **94.91%** |
| CIFAR-100 (Top-1 acc.) | 70.48% | 71.02% | 73.31% | 73.12% | 73.81% | **74.94%** |
| CIFAR-100 (Top-5 acc.) | 92.16% | 92.23% | 93.51% | 92.31% | 92.37% | **93.66%** |
| Part **III** (image data) | Res101 | Ens-Res101 | DEns-Res101 | Res152 | Ens-Res152 | DEns-Res152 |
| CIFAR-10 (training time) | 12m | 45m | 15m | 19m | 1h 13m | 24m |
| CIFAR-100 (training time) | 27m | 1h 40m | 31m | 43m | 2h 21m | 49m |
| CIFAR-10 (model size) | 43.3M | 136.1M | 44.7M | 60.7M | 231.8M | 62.3M |

Table 3: Sentiment classification accuracy (%) on six text datasets.

| Method | 20NG | R8 | R52 | Ohsumed | MR | SST |
|---|---|---|---|---|---|---|
| TF-IDF + LR | 83.19 | 93.74 | 86.95 | 54.66 | 74.59 | 41.59 |
| TF-IDF + SVM | 82.27 | 94.03 | 87.83 | 53.17 | 75.58 | 42.57 |
| CNN-rand | 76.93 | 94.02 | 85.37 | 43.87 | 74.98 | 45.29 |
| DEns-CNN-rand | 78.63 | 95.01 | 86.43 | 44.12 | 75.53 | 46.39 |
| CNN-non-static | 82.15 | 95.71 | 87.59 | 58.44 | 77.75 | 49.32 |
| DEns-CNN-non-static | 83.37 | 96.84 | 87.73 | 60.70 | 78.52 | 50.33 |
| GRU | 73.62 | 91.44 | 84.58 | 49.92 | 76.39 | 47.03 |
| DEns-GRU | 75.70 | 92.01 | 85.56 | 50.38 | 77.03 | 49.45 |
| LSTM | 65.71 | 93.68 | 85.54 | 41.13 | 75.06 | 48.85 |
| DEns-LSTM | 69.57 | 95.36 | 87.07 | 47.93 | 77.40 | 49.53 |
| BiLSTM | 73.18 | 96.31 | 90.54 | 49.27 | 77.68 | 49.72 |
| DEns-BiLSTM | 77.30 | 97.09 | 92.15 | 51.29 | **79.34** | 51.19 |
| Graph-CNN | 81.58 | 96.67 | 92.54 | 62.19 | 76.94 | 50.86 |
| DEns-Graph-CNN | 82.29 | 97.14 | 93.34 | 62.97 | 79.19 | **52.28** |
| Text-GCN | 85.93 | 97.01 | 93.67 | 67.94 | 77.14 | 50.03 |
| DEns-Text-GCN | **87.06** | **97.73** | **94.29** | **68.21** | 78.37 | 51.37 |

Table 4: Accuracy and MAE comparison with different number of used modules i.e. $N$.

| Datasets | $N = 1$ | $N = 2$ | $N = 3$ | $N = 4$ | $N = 5$ | $N = 6$ |
|---|---|---|---|---|---|---|
| Higgs Boson | 82.83% | 82.95% | 82.97% | **83.08%** | 83.01% | 82.99% |
| KDD10 | 77.14% | 78.51% | 78.87% | **79.19%** | 79.02% | 78.76% |
| CIFAR-10 (Top-1) | 93.56% | 94.01% | 94.32% | **94.89%** | 94.81% | 94.66% |
| CIFAR-100 (Top-1) | 70.48 % | 72.26 % | 72.91% | **73.31%** | 72.98% | 72.67% |
| Credit (MAE) | 14.9901 | 14.6741 | 14.3318 | 14.1103 | **14.1028** | 14.2881 |

Part III of Table 2 shows the time and space comparison on the CIFAR-10 and CIFAR-100 datasets. Compared with the traditional aggregating method, the additional parameters require to estimate are reduced from $\#hidden \times \#node \times \#model + \#model$ to $\#node \times \#model + \#model$. It can be seen that the additional time spent on ResNet-152 is more than on ResNet-101. The reason is that the extra time required for each back-propagation of the ensemble layer in the deeper network is more. At the same time, due to the reason of weight sharing, the extra space required by the model is also greatly reduced. Only one more linear layer of space is needed to complete the efficient ensemble.

**Results on text datasets.** Table 3 shows the accuracy score on text datasets. In *CNN-rand* model, all the word vectors are initialized randomly before training, while *CNN-non-static* adopts pre-trained word2vec to embed words, which are fixed in the training process. Table 2 and Table 3 indicate with our ensemble layers, RNNs, CNNs, and GCNs mostly outperform.

## 4.3 Further Study

We perform additional ablation study to our method as follows.

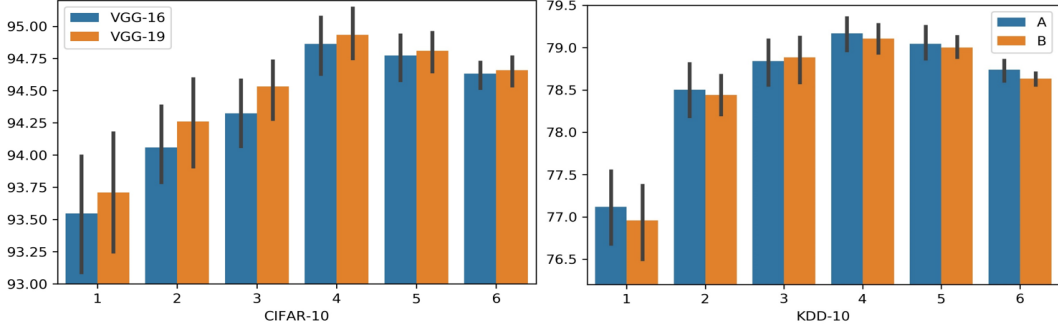

Figure 2: Accuracy (%) by different $N$: (difference of lowest and mean, mean, difference of highest and mean in 10 trials). For CIFAR-10 (left), we adopt VGGs as backbones. While on the right plot, A and B in KDD dataset (right) mean two different fully connected networks.

Table 5: Accuracy (%) on E-Text-GCN with different values of $\alpha$ including the computed ones by our Sigmod function. Note for $\alpha = 0$, it indicates the diversity loss is removed.

| dataset | $\alpha = 0$ | $\alpha = 0.2$ | $\alpha = 0.4$ | $\alpha = 0.6$ | $\alpha = 0.8$ | $\alpha = 1$ | $\sigma(\mathbb{E}_{\mathbf{W}}(|\nabla_{\mathbf{W}}\mathcal{L}_s^{(i-1)}|))$ |
|---|---|---|---|---|---|---|---|
| 20NG | 86.01 | 86.53 | 86.71 | 86.69 | 85.51 | 85.02 | **87.06** |
| R8 | 97.07 | 97.41 | 97.39 | 97.34 | 96.37 | 95.39 | **97.73** |
| R52 | 93.82 | 94.07 | 94.13 | 94.11 | 93.91 | 92.79 | **94.29** |
| Ohsumed | 68.06 | 68.09 | **68.23** | 68.14 | 67.79 | 67.01 | 68.21 |
| MR | 77.31 | 77.68 | 77.91 | 78.03 | 77.28 | 70.75 | **78.37** |
| SST | 50.19 | 50.73 | 51.04 | 50.91 | 50.33 | 49.38 | **51.37** |

i) To find the best $N$-value, we conduct additional experiments on the tabular dataset and image dataset. Specifically, we change the number of tiers in the ensemble layer with the same network structure. Part V in Table 4 shows the result. On CIFAR-10 and CIFAR-100, we adopt VGG-16 as the CNN structure. When $N = 1$, it causes $\mathcal{L}_d = 0$, which is the vanilla CNN. When $N = 4$, the average accuracy on five datasets is the best. And with the increase of $N$, the extra time for training is gradually increasing. When $N >> \#hidden$, the time complexity of traditional aggregation and ensemble layer are both $O(N)$.

ii) In the ensemble layer, $\alpha$ is employed to balance the attenuation factor between two $\mathcal{L}_s$ and $\mathcal{L}_d$. When $\alpha = 0$, the diversity between the individual models is unstable, which can be greatly influenced by weight initiation; when $\alpha$ is too large, the individual models can not reach the optimum. Since in the training process, the diversity loss $\mathcal{L}_d$ between the models will gradually increase, while the loss $\mathcal{L}_s$ will decrease progressively. Therefore, when $\alpha$ is a non-zero constant, the individual model will be unstable. We compare the accuracy of different $\alpha$ and the experimental results are given in Table 5. We can observe that our adaptive setting technique for $\alpha$ outperforms those with a fixed one notably.

**Ensemble variance.** Ensemble is a general way to reduce the variance and improve generalization. To verify this, we conduct extensive experiments on tabular dataset and image dataset. Fig. 2 shows accuracy variation by increasing $N$. When $N$ gets larger, the accuracy fluctuation becomes smaller.

# 5 Conclusion

In this paper, we propose the so-called ensemble layer as a building block for combining multiple deep neural networks, such that the training error $\mathcal{L}_s$ is minimized while the individual diversity $\mathcal{L}_d$ can be maximized. Moreover, we theoretically prove the generalization and accuracy improvement by our ensemble technique. Extensive experimental results show that our ensemble layer greatly improves the accuracy on training and test datasets with negligible extra time and space cost. The proposed ensemble layer can also be portable to popular networks such as GCNs, RNNs, and CNNs.

## Broader Impact

Ensemble is a general technology to improve the performance of machine learning models. This paper makes contributions to ensemble technology, which may ultimately improve the performance of AI systems. The potential risk is the possible formation of super AI out of the control of human beings. Also, individual privacy may be put at risk due to the strengthened AI capability.

## Acknowledgments and Disclosure of Funding

The work is partially supported by National Key Research and Development Program of China, No. 2020AAA0107600 and NSFC 61972250, U19B2035, 72061127003, and CCF-Tencent Open Fund RAGR20200113 and Tencent AI Lab Rhino-Bird Visiting Scholars Program.

## Footnotes

*Junchi Yan is the correspondence author.

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
