[Reviews · NeurIPS 2020]

Review 1

Summary and Contributions: This paper proposes a principled ensemble technique by constructing the so-called diversified ensemble layer to combine multiple networks as individual modules. The authors have theoretically shown that each individual model in the ensemble layer corresponds to weights in the ensemble layer optimized in different directions.

Strengths: It is an interesting idea to explore ensemble learning with neural networks. The authors have done some insightful analysis to show that the diversity constraints benefits for improving the genratlization. The experimental results are comprehensive to justify the effectivenss of the proposed algorithm.

Weaknesses: In Eqs. (3) and (4), the authors explored the diversity constraint. It seems that the authors aim to maxize the difference between the feature maps. More explanations are expected here. How the features map difference would be necessary and beneficial for the the whole model performance improvement. In ensemble learning, it is expected that the weak learners together would lead to a strong learner. It is an interesting idea to explore ensemble learning over neura networks. But will each individual module need to be weak? If each module is already strong, then integrating them further may cost a lot but bring in only some limited performance improvement.

Correctness: Most of the claims are correct. It would be great to further explain the necessity of exploring ensemble learning with nerual networks.

Clarity: The paper has been well written.

Relation to Prior Work: The authors have clearly discussed related works in the paper.

Reproducibility: Yes

Additional Feedback: Post rebuttal: The authors have well addressed my concerns. I would like to recommend its acceptance.


Review 2

Summary and Contributions: The authors present a new neural network layer they call the "ensemble layer" that can be appended to existing feature extractors (such as CNNs, RNNs and GCNs) and trained jointly with them, which provides a diversity of predictions leading to higher accuracy when ensembled together. The authors provide a theoretical proof of the improvement, show an extensive testing on a range of modalities and architectures, and demonstrate the relatively low memory and time complexity of their proposed solution.

Strengths: I really appreciate the extensive testing the authors have done with a variety of modalities (text, images, graphs) as well as architectures. I also like that they made a good comparison in terms of time and memory complexity of their method as compared to others. The idea seems sound and follows a line of similarly spirited research where only the latter parts of a NN / additional structure on top of a feature extractor NN are trained to obtain some diversity of predictions on a task. I also appreciate the mathematical proofs that they presented that complement the empirical analysis. Overall I think the paper has a good balance of experimental and theoretical work. In terms of relevance I think there is a great need to methods that do get a partially to where plain ensembles would go, but at a much lower cost, which is what the authors present here.

Weaknesses: 1) I would like to see errorbars on the numbers the authors present. While this is not common universally common ML, I'd really like to see errorbars on the numbers the authors are presenting. When showing that e.g. an ensemble of a bigger size is worse than of a smaller size (that can certainly happen for a specific set of models that we ensemble in some order), the trend might not be well captured by a single set of numbers. Seeing a mean +- error would be good for distinguishing whether some minute details of accuracy (such as on CIFAR-10, where we are well above 90% test accuracy) are due to luck, or a systematic improvement / different due to the method. This could really benefit e.g. Table 3, where the improvements often are not monotonic in N. 2) Sharing a feature extractor = a hard limitation on how different the fns can be? I understand that sharing of the feature extractor (such as the CNN / RNN / GCN in Figure 1) is needed to save parameters and training time, and of course the gain in diversity of functions would have to be bought by partially sacrificing those. I wonder if using e.g. the pretrained VGGs or ResNets (as mentioned in Section 3.1) already limits how diverse the resulting functions on top of those you can get. This is not a point of criticism, but rather a thought. It would be great to see this addressed explicitly in the work, but not having it is not a great detriment to the overall quality of the paper.

Correctness: The claims seem correct and so does the methodology.

Clarity: The paper is clearly written.

Relation to Prior Work: While I am not 100% familiar with the relevant literature, one paper that seems relevant in spirit is *BatchEnsemble: An Alternative Approach to Efficient Ensemble and Lifelong Learning* (https://arxiv.org/abs/2002.06715) where they also apply a technique that does not provide the full diversity of an ensemble of N independent models, but goes partially there while saving a lot of parameters and compute. The diversity of predictions and its tradeoff with accuracy was also studied in detail in *Deep Ensembles: A Loss Landscape Perspective* https://arxiv.org/abs/1912.02757 which might be relevant as well.

Reproducibility: Yes

Additional Feedback: POST REBUTTAL: The authors addressed my points well and I'm keeping my score at 7 = accept. I think that this paper is a valuable addition to the field.


Review 3

Summary and Contributions: The authors propose a new ensemble method named the diversified ensemble neural network, which diversifies the last several layer outputs from multiple individual networks to boost the ensemble performance.

Strengths: Overall, I think this paper is relatively clear and necessary theorems and proofs have been provided. What's more, the authors run a number of experiments, which demonstrate that the proposed method shows competitive performance in a variety of tasks.

Weaknesses: The related work section is relatively weak. The authors only compare the proposed ensemble method with classic algorithms such as XGBoost with ignoring some state-of-the-art ensemble diversity enhancement methods. For instance, Zhao W, Zheng B, Lin Q, et al. Enhancing diversity of defocus blur detectors via cross-ensemble network[C]//Proceedings of the IEEE Conference on Computer Vision and Pattern Recognition. 2019: 8905-8913. Pang T, Xu K, Du C, et al. Improving Adversarial Robustness via Promoting Ensemble Diversity[C]//International Conference on Machine Learning. 2019: 4970-4979. The authors failed to discuss such similar work, which designs similar ensemble architectures aiming for diversifying the outputs from multiple individual neural networks. Therefore, the contribution of this paper seems limited.

Correctness: The paper is technically sound.

Clarity: The paper is generally well written and is a pleasure to read.

Relation to Prior Work: Details are in the weakness section.

Reproducibility: No

Additional Feedback: Update: We still suggest the authors should compare the proposed ensemble method with more ensemble strategies, especially for those methods prepared for diversifying the features in the last few layers. So, I keep my score as before.


Review 4

Summary and Contributions: This paper proposes an algorithm to induce diversity in NN framework by introducing a diversified layer; this layer is trained on top of some shared layers being trained on the output of any kind of NN architecture. The authors provide some proof on why this layer can help and it does not add much overhead to NN training. Extensive sets of experiments are also provided to show the effectiveness of this method

Strengths: - The proposed method does not add much overhead computationally. - The idea is novel, although some relevant works could extend the references. - The paper provides theoretical proof on the idea

Weaknesses: - Few extra experiments are required to make sure the improvements are explainable. (please refer to the additional feedback)

Correctness: Yes

Clarity: The paper is well written. Some extra explanation regarding hyper parameter tuning and diversified layer could help. (please refer to the additional feedback)

Relation to Prior Work: To a good extent

Reproducibility: No

Additional Feedback: 1- does each module in the diversified ensemble layer consist of only one linear layer or a deep NN again? From formula 3 it looks like it is only a linear layer, if so, have the authors considered more layers and how would they argue to use one layer versus more. Also, the current loss function for diversity would still be valid in the case of “deep diversified ensemble layer”? 2- in the abstract, it is rightly argued that generalization on small datasets is challenging; however, in the experiments it is not clear if any is done particularly towards this goal. In other words, I am wondering how we can make sure the improvements presented by using DEns-NN contributes to generalization? One possible suggestion could be tracking L_d (diversity loss function) during training on both training data and some validation data; it can be insightful to see how much of error reduction is correlated/due to L_d. 3- In the same line, I am also wondering how the hyperparameters in R-Forest, XGBoost, NN are set? Are they different across different datasets? Are they set individually for each dataset via a validation set? 4- The paper is written clearly; however, it can improve by revisiting the use of notations; for example N is used for different purposes. Also, there are some typos. 5- While the proposed algorithm is novel to my best knowledge, there are certain works in literature which attempt towards the same goal. For example, in [1] diversity is induce in different NNs and parameters are combined by learnable weights; therefore, like this work it benefits from parameter sharing and combination weights. Or the ROVER system [2] in speech recognition benefits from some voting module which works similar to aggregate weight in this work. It can be helpful to extend your bibliography with this sort of references. [1] Cross-entropy training of DNN ensemble acoustic models for low-resource ASR [2] A Post-processing System to Yield Reduced Word Error Rates: Recongiser Output Voting Error Reduction(ROVER) ------------ After rebuttal: I appreciate the authors' effort in responding the comments.

[Author Response · NeurIPS 2020]

**Reviewer #1**

**Maximizing feature map difference?** Thanks for your question. The Eqs. (3) and (4) do relate to the diversity maximization among individual learners. While here we clarify that our diversity maximization mechanism has nothing to the feature map part. The feature map or say the extracted features from the raw input data is shared by all the individual learners as their input. We will emphasize this fact in our final version.

**Individual model must be weak? and the necessity of ensemble for NN?** Thanks for your question which gives us good opportunity to further clarify our motivation and the position of this paper. We note that one possible reason for the lack use of neural networks for ensemble in the community may be due to their strong fitting ability. This on one hand reduces the need for ensemble and on the other hand, the ensemble can in turn cause over-fitting. In this paper, we address this issue in two folds: i) we introduce weight sharing to control (reduce) the model capacity of each network leaner; ii) we introduce the diversity loss to mitigate the over-fitting and enhance the generalization ability.

Our approach allows some space to control the complexity of each leaner and the generalization ability on new dataset. Considering the readily available network modules and fast development in this area, it opens more possibility for further research opportunities. The applicability can also be justified considering the use of the GPU power.

**Reviewer #2**

**Errorbar.** The following table shows accuracy variation by increasing $N$. When $N$ gets larger, the accuracy fluctuation becomes smaller. The accuracy in Table 3 in our paper is the mean of 10 trials. We will add errorbars in final version.

| dataset | $N$=1 | $N$=2 | $N$=3 | $N$=4 | $N$=5 | $N$=6 |
|---|---|---|---|---|---|---|
| KDD10 | -0.46 **77.14** +0.40 | -0.31 **78.51** +0.30 | -0.31 **78.87** +0.22 | -0.22 **79.19** +0.16 | -0.15 **79.02** +0.23 | -0.15 **78.76** +0.09 |
| Cifar-10 | -0.47 **93.56** +0.43 | -0.23 **94.01** +0.37 | -0.25 **94.32** +0.26 | -0.21 **94.89** +0.18 | -0.16 **94.81** +0.12 | -0.14 **94.66** +0.08 |

Table 1: Accuracy (%) by different $N$: (difference of lowest and mean, mean, difference of highest and mean).

**Will pre-trained VGG and Resnet limit the diversity?** Note that the diversity is fulfilled by the learned individual learners (i.e. their own parameters) **after the shared layers**. VGG or Resnet can be regarded as part of the weight sharing parts hence weather it is pre-trained or random initialized has nothing to do with our diversity maximization mechanism. In another word, the pretrained models serve as a feature extractor which can be application dependent.

**Reviewer #3**

**Difference to the two previous works.** The two mentioned papers are both application dependent whose design are much coupled with the specific application: defocus blur detection and network robustifying.

1) Difference to DBD-CENet: **i) Methodology.** The mentioned paper DBD-CENet is more like a boosting ensemble strategy, which also combines the idea of knowledge distillation (teacher-student network) and co-training. While our ensemble layer is more like blending and bagging method. **ii) Techniques.** DBD-CENet uses one branch to estimate the residual of the other branch iteratively. Finally, they finetune the two branches of DBD-CENet in an iterative way with every epoch (mentioned in Eq. (9) and Eq. (10) in that paper). These adhoc practices are all not used in our work.

2) Difference to adaptive diversity promoting (ADP): **i) Methodology.** ADP's diversity is fulfilled by dot product in embedding layer. Hence it is difficult to be applied on tabular data sets and regression task. While our ensemble layer gives a more universal approach to quantify diversity, and it can be trivially applied to these tasks. Besides, since ADP directly use original data as input to ensemble, its time and space overhead can increase exponentially, which also limits its applicability. **ii) Techniques.** One notable difference for handling the diversity and accuracy loss is that ADP keeps the weights of the two loss constant (see in Eq. (5) in that paper) while in our method the weights of the two loss change over time for more effective learning. We will discuss the differences more detailedly in our final version.

**Reviewer #4**

**Number of layers for the ensemble layer?** In all our experiments, we only use one layer as the ensemble layer, and we once have tested by increasing the number of its layers while no performance improvement is observed. This may suggest that the diversity can be readily realized by a moderate sized model i.e one layer network, and a more complex model can even hurt the performance. We will add comparison and discussion in final version by the 9th extra page.

**Generalization test?** The numbers for training and testing loss curve will be added in our final version.

**Compared methods tuning?** Yes, all the methods are finetuned to the best performance according to a validation set, such as max_depth, num_leaves, learning_rate, etc. This will be clarified in the final version.

**Relevant literature.** Thanks and the mentioned papers will be added in final version. Their difference will be discussed.

[Meta-Review · NeurIPS 2020]

Two referees support accept, one weak-accept and one indicates reject. Rebuttal clarified R3’s concerns regarding two related work directions, but R3 did not respond during the reviewer discussion. R4 (weak-accept) did engage and was happy with the rebuttal connected with their concerns. R1 and R2 were happier with their accept stance post-rebuttal. I therefore think that this paper should be accepted. However, please consider closely proof-reading your paper for clarity and correctness.